# Formation of Amino Acids and Carboxylic Acids in Weakly Reducing Planetary Atmospheres by Solar Energetic Particles from the Young Sun

**DOI:** 10.3390/life13051103

**Published:** 2023-04-28

**Authors:** Kensei Kobayashi, Jun-ichi Ise, Ryohei Aoki, Miei Kinoshita, Koki Naito, Takumi Udo, Bhagawati Kunwar, Jun-ichi Takahashi, Hiromi Shibata, Hajime Mita, Hitoshi Fukuda, Yoshiyuki Oguri, Kimitaka Kawamura, Yoko Kebukawa, Vladimir S. Airapetian

**Affiliations:** 1Department of Chemistry and Life Science, Graduate School of Engineering Science, Yokohama National University, 79-5 Tokiwadai, Hodogaya-ku, Yokohama 240-8501, Japan; 2Chubu Institute of Advanced Studies, Chubu University, 1200 Matsumoto-cho, Kasugai 487-8501, Japan; 3SANKEN (The Institute of Scientific and Industrial Research), Osaka University, Ibaraki 567-0047, Japan; 4Department of Life, Environment and Applied Chemistry, Faculty of Engineering, Fukuoka Institute of Technology, Fukuoka 811-0295, Japan; 5Open Facility Center, Tokyo Institute of Technology, Meguro-ku, Tokyo 152-8550, Japan; 6Institute of Innovative Research, Tokyo Institute of Technology, Tokyo Institute of Technology, Meguro-ku, Tokyo 152-8550, Japan; 7NASA Goddard Space Flight Center/Sellers Exoplanetary Environments Collaboration, Greenbelt, MD 20771, USA; 8Department of Physics, American University, Washington, DC 20016, USA; 9Graduate School of Advanced Integrated Studies in Human Survivability, Kyoto University, Sakyo-ku, Kyoto 606-8501, Japan

**Keywords:** prebiotic synthesis, cosmic rays, solar energetic particles, weakly reducing atmosphere, early Earth, amino acids, carboxylic acids

## Abstract

Life most likely started during the Hadean Eon; however, the environmental conditions which contributed to the complexity of its chemistry are poorly known. A better understanding of various environmental conditions, including global (heliospheric) and local (atmospheric, surface, and oceanic), along with the internal dynamic conditions of the early Earth, are required to understand the onset of abiogenesis. Herein, we examine the contributions of galactic cosmic rays (GCRs) and solar energetic particles (SEPs) associated with superflares from the young Sun to the formation of amino acids and carboxylic acids in weakly reduced gas mixtures representing the early Earth’s atmosphere. We also compare the products with those introduced by lightning events and solar ultraviolet light (UV). In a series of laboratory experiments, we detected and characterized the formation of amino acids and carboxylic acids via proton irradiation of a mixture of carbon dioxide, methane, nitrogen, and water in various mixing ratios. These experiments show the detection of amino acids after acid hydrolysis when 0.5% (*v*/*v*) of initial methane was introduced to the gas mixture. In the set of experiments with spark discharges (simulation of lightning flashes) performed for the same gas mixture, we found that at least 15% methane was required to detect the formation of amino acids, and no amino acids were detected in experiments via UV irradiation, even when 50% methane was used. Carboxylic acids were formed in non-reducing gas mixtures (0% methane) by proton irradiation and spark discharges. Hence, we suggest that GCRs and SEP events from the young Sun represent the most effective energy sources for the prebiotic formation of biologically important organic compounds from weakly reducing atmospheres. Since the energy flux of space weather, which generated frequent SEPs from the young Sun in the first 600 million years after the birth of the solar system, was expected to be much greater than that of GCRs, we conclude that SEP-driven energetic protons are the most promising energy sources for the prebiotic production of bioorganic compounds in the atmosphere of the Hadean Earth.

## 1. Introduction

The origin of life on Earth during the Hadean period is one of the fundamental puzzles of modern science. To understand its chemical complexity, we need to examine the global space weather conditions imposed by magnetically driven eruptive events from the young Sun and the surface, atmospheric, and oceanic environments of the early Earth, which contributed to the onset of the synthesis of complex biopolymers on the surface of our planet.

Geophysical and fossil data provide evidence that life arose 4.5–3.8 billion years ago on the Hadean Earth [1,2]. The early Haldane–Oparin theory of the origin of life suggested that first abiogenic molecules may have been driven by chemical disequilibrium via external energy sources in a strongly reducing atmosphere with an abundance of methane, CH_4_, ammonia, NH_3_, and water (H_2_O) [3,4]. This theory was tested by the famous Miller–Urey experiments, in which spark discharge was employed as a source of energy to generate amino acids and carboxylic acids in such a strongly reduced gas mixture [5]. The Miller–Urey experiments were followed by other studies using energy sources such as heat [6], ultraviolet light (UV) [7], ionizing radiation [8], and shock waves [9] to produce amino acids and other prebiotic molecules. However, later photochemical studies suggested that ammonia or methane would have been quickly destroyed by UV emissions from the young Sun [10,11]. Moreover, geochemical data supported the arguments in favor of a weakly reducing N_2_–CO_2_-dominated atmosphere driven by volcanic outgassing from the solid Earth [12,13,14,15]. Miller–Urey-type experiments performed with spark discharges and UV irradiation of such oxidized mixtures did not show efficient production of prebiotically important molecules [16,17,18,19]. Specifically, the reduced production rates driven by UV emission in a weakly reduced atmosphere are caused by the inefficiency of forming N-containing organics, since the photodissociation of N≡N bonds requires extreme UV emissions at wavelengths < 110 nm [20].

Galactic cosmic rays (GCRs) represent another possible energy source for the prebiotic formation of organic compounds in the early Earth’s atmosphere. To simulate the action of GCRs in early planetary atmospheres, Kobayashi et al. performed proton irradiation experiments in CH_4_ (or CO), N_2_, and H_2_O-containing gas mixtures and reported the production of high yields of amino acids [21,22], in contrast to significantly smaller yields of amino acids in CO_2_, N_2_, and H_2_O mixtures [21].

Miller and Urey [16] argued that the contribution of the GCR energy flux penetrating the Earth’s atmosphere to prebiotic synthesis was negligible as compared to other energysources, such as solar UV and lightning events. Using the updated GCR spectrum reported by Meyer et al. (1974) [23], Kobayashi et al. derived the particle intensity of 0.011 cm^−2^ yr^−1^ (2π)^−1^ [22]. This suggested that GCRs may represent a viable energy source to produce amino acids in CO_2_-N_2_-rich atmospheres, since the energy yield of amino acids via high-energy protons is much higher than those provided by lightning flashes (or spark discharges) or by UV photons [22,24]. However, the GCR intensity around the Earth during the Hadean period was estimated to be at least two orders of magnitude lower than that observed today [25]. This intensity reduction was caused by the faster rotation of the young Sun, which wound up its strong magnetic field carried by the solar wind, and thus increased the azimuthal component of the interplanetary magnetic field. The stronger magnetic field then efficiently swept GCR particles (with energy less than 10 GeV) away, and thus significantly reduced the penetration flux into the Earth’s atmosphere. This process contributes to the modulation of GCR flux over the solar cycle to a lesser degree when the global magnetic field of the current Sun undergoes restructuring, which is known as the Forbush effect [26]. While stronger magnetic fields from the young Sun weakened the flux of GCRs into the Earth’s atmosphere, the solar magnetic activity fueled the production of frequent and energetic superflares, with energy up to 1000 times greater than in flares observed in the current Sun [27,28]. As observed in young solar-like stars, including 100-Myr-old EK Dra, superflares are accompanied by filament and prominence eruptions that are associated with the ejection of massive and fast magnetized clouds, referred to as coronal mass ejections, or CMEs [29,30,31,32]. Recently, Hu et al. simulated the production of solar energetic particles (SEPs) associated with superflares and CMEs from young solar-like stars representing our Sun at ages of 100 and 600 Myrs [32]. These theoretical models appear to be consistent with the extrapolation of the empirical scaling between solar flare peak intensity and SEP particle flux to superflares from young solar-like stars. This suggests that high-flux, hard-spectra SEPs were generated from the young 100-Myr-old Sun every few days and penetrated the Earth’s atmosphere, which would have been an important energy source for the initiation of prebiotic chemistry [33]. The energy fluxes to the atmosphere of the Hadean Earth supplied by such energetic SEP events were over 7 orders of magnitude greater than those provided by GCRs [32]. This suggests that SEPs from the young Sun could be considered as another possible energy source for prebiotic synthesis [34].

While it is usually assumed that the early Earth’s atmosphere was weakly reducing, its exact composition is not well understood [35]. A recent study suggested that giant impacts during the Hadean period could have contributed to a transient methane-generated atmosphere lasting for about 15 million years [36]. Together with the high carbon fraction, methane could be an important factor in the early Earth atmosphere to boost the production of biologically relevant molecules, thus boosting the production of biologically relevant molecules [37]. For example, amino acid precursors were effectively formed in a mixture of N_2_ (95%) and CH_4_ (5%) by proton irradiation, simulating reactions in the lower atmosphere of Titan [38]. In contrast, CO_2_, which appears to have been the major carbon species in the early Earth’s atmosphere, would not have been a good starting material for prebiotic synthesis [39] unless a large amount of reducing species were coexisting [17], and might have inhibited the formation of amino acids from N_2_ and CH_4_. The ratio of CH_4_ to CO_2_ in the early Earth’s atmosphere was estimated to be 0.13–7.6% [40].

In the present study, we examined the formation of amino acids and carboxylic acids in simulated early Earth atmospheres of a weakly reducing type, containing a mixture of N_2_, CO_2_, CH_4_, and H_2_O in various mixing ratios. We characterized the products of the impact of spark discharges (a proxy for lightning events), proton irradiation (proxies for GCRs/SEPs), and solar UV radiation as energy sources of prebiotic chemistry.

## 2. Materials and Methods

### 2.1. Chemicals

The gases used in our experiments included (i) 50.0: 50.0 (*v*/*v*) mixture of CH_4_ and N_2_ (Toho Sanso Co., Japan); (ii) 1.02: 98.98 (*v*/*v*) mixture of CH_4_ and N_2_ (Taiyo Nippon Sanso Co., Japan); (iii) 50.0: 50.0 (*v*/*v*) mixture of CO_2_ and N_2_ (Taiyo Nippon Sanso Co., Japan); (iv) CH_4_ (Taiyo Nippon Sanso Co., Japan); (v) CO_2_ (Suzuki Shokan Co., Japan); and (vi) N_2_ (Toho Sanso Co., Japan). Gas mixtures of CO_2_, CH_4_, and N_2_ in various mixing ratios were prepared by mixing the purchased gases listed above.

Standard amino acid mixture solutions (Type AN-II and B; Fujifilm Wako Pure Chemicals Co., Japan) were used to the determine amino acids in the products. HCl (amino acid analysis grade) was purchased from Fujifilm Wako Pure Chemicals Co., Japan. Water was purified with a Milli-Q system. All the glassware and metallic parts used were heated in an electric oven at 500 °C before use for the sterilization and removal of organic compounds.

### 2.2. Instruments

The cation-exchange HPLC used was a Shimadzu Amino Acid Analysis System including two LC-10AT pumps, a RF-20Axs fluorescence detector, and a Shimpak ISC-07/S 1504 column (4.0 mm i.d. × 150 mm). Amino acids in effluents were derivatized with o-phthalaldehyde and N-acetyl-L-cysteine for fluorometric detection. Details of the system were described in previous papers [41,42]. Amino acids were also analyzed with a gas chromatograph mass spectrometer (a Shimadzu GCMS-QP2020; column: an Agilent CP-Chirasil Val, 25 m long × 0.25 mm i.d. × 0.12 mm film thickness, with which D- and L-amino acid isomers could be separated).

Two types of carboxylic acids (monocarboxylic and dicarboxylic acids) were determined using a capillary gas chromatograph (GC, Agilent model 6890) equipped with a split/splitless injector, fused silica capillary column (DB-5, 25 m long × 0.25 mm i.d. × 0.5 mm film thickness), and flame ionization detector (FID). The structural identification of the esters was confirmed by GC/mass spectrometer (GC/MS, Agilent model 6890 GC, and Agilent model 5975 mass selective detector).

### 2.3. Proton Irradiation

We introduced a gas mixture of N_2_, CO_2_, and CH_4_ in a Pyrex glass tube (400 mL) with a Havar foil window (purchased from Nilaco Co., Japan) [38] and 5 mL of pure water. Then, the gas mixture was irradiated with a 2.5 MeV proton beam from a Tandem accelerator (Tokyo Institute of Technology, Japan) at ambient temperature. The beam size was approximately 5 mm when it passed through the window (Havar foil) of the tube. The composition of the starting gas mixtures is shown in Table 1. The partial pressure of water (*p*H_2_O) in the gas mixtures was kept fixed at about 20 Torr (vapor pressure of water at 22 °C), which is not shown in the table.

Here, the CH_4_ ratio (*r*CH_4_) was defined as *p*CH_4_/(*p*CH_4_ + *p*CO_2_ + *p*N_2_). The energy of the proton decreased to 1.58 MeV after passing the Havar foil, and the current was ca. 0.5 μA. The proton irradiation time was about 1–2 h, during which the total quantity of protons was controlled at 2 mC. The total energy deposit to each gas mixture was estimated to be 3.16 kJ. This corresponded to the input proton intensity of 1.97 × 10^18^ eV cm^−2^ s^−1^. The experimental input proton flux is comparable with the estimation of the proton flux at E > 10 MeV, associated with an energetic superflare based on the size distribution of solar and stellar flares [33]. This scaling suggests that the peak intensity of StEPs associated with superflare events with energy ~3 × 10^34^ erg was ~10^18^ eV cm^−2^ s^−1^, and, thus, was of the same order as the input proton intensity used in our proton irradiation experiments.

### 2.4. Spark Discharge

A gas mixture of N_2_, CO_2_, and CH_4_ was introduced in a Pyrex glass flask with 2 tungsten electrodes and 5 mL of water. The composition of the gas mixture is shown in Table 1. Details of the apparatus are shown in [39]. Spark discharges were performed using a Tesla coil (Electro-Technic Products BD-50) for 24 h with a 50% duty cycle (1 min on/1 min off). The total energy deposit to the gas mixture was estimated to be 864 kJ by the oscilloscopic measurement [43].

### 2.5. Analysis of the Products

After spark discharge/proton irradiation, the resulting gaseous product was taken from the flask/tube and analyzed by GC/MS. Then, the aqueous phase product was recovered and the vessel was rinsed with 5 mL of pure water. The wash water was added to the original product.

An aliquot of the product was acid-hydrolyzed at 110 °C for 24 h in 6 M HCl; then, HCl was removed by vacuum centrifugation. Both the products with and without acid-hydrolysis were analyzed by ion-exchange HPLC to determine the amino acids. Details of the amino acid analysis are shown in [44]. Some of the hydrolyzed samples were also analyzed by GC/MS after derivatization with 2, 2, 3, 3, 4, 4, 4-heptafluoro-1-butanol and ethylchloroformate [38].

Another aliquot of the product was subjected to alkaline hydrolysis at 95 °C for 1.5 h in 0.05 M KOH. Both products with and without alkaline hydrolysis were derivatized to *p*-bromophenacyl esters for monocarboxylic acids (C_1_ to C_10_) [45] and dibutyl esters for dicarboxylic acids (C_2_ to C_10_) [46], and then determined by GC/FID and GC/MS, as stated above.

## 3. Results

### 3.1. Formation of Amino Acids

Figure 1 shows typical HPLC chromatograms of the amino acids formed by spark discharges. A chromatogram of standard amino acids is shown in Appendix A. All the samples were acid-hydrolyzed before analysis. When *r*CH_4_ was 0.2 or over, various amino acids were detected. Glycine (Gly) was predominant, followed by amino acids such as aspartic acid (Asp), serine (Ser), a-aminobutyric acid (a-ABA), and b-alanine (b-Ala). The alanine (Ala) peak was sometimes not identified since its peak appeared in the tail of a larger Gly peak, but Ala could be detected by GC/MS: Ala in the products was a racemic mixture, which showed that it was indigenous, not contaminated. On the other hand, amino acids were not detected when *r*CH_4_ was 0.05 or less. Samples without hydrolysis showed only trace levels of amino acids independently on *r*CH_4_ ratios. These results showed that not free amino acids, but amino acid precursors, were formed when the high partial pressure of methane was applied in CO_2_-CH_4_-N_2_ type atmospheres.

Figure 2 shows HPLC chromatograms of amino acids in the hydrolyzed samples formed by proton irradiation. Here, no amino acids were detected in unhydrolyzed samples. In the case of proton irradiation, amino acid precursors could be formed even when the gas mixture was weakly reducing with quite low *r*CH_4_. In all the cases, Gly was predominant, and α-ABA and β-Ala followed. The Ala peak could not be separated from the Gly peak for the same reason as in the case of spark discharges.

In Figure 3 and Figure 4, the yields of major amino acids determined by HPLC are plotted against the starting methane ratios. Open symbols specify non-detected amounts of amino acids (ND). In the case of spark discharges (Figure 3), no amino acids were found at *r*CH_4_ = 0–0.1, while trace levels of amino acids were detected at *r*CH_4_ = 0.15, and amino acid yields drastically enhanced at *r*CH_4_ > 0.2. The yield of α-amino acids such as Gly and α-ABA increased with the *r*CH_4_ value, while the yield of β-alanine (β-amino acid) decreased at an *r*CH_4_ greater than 0.3. 

In the proton irradiation experiments, however, amino acids were formed even under the low (0.005–0.05) *r*CH_4_ condition, where the yields increased with the *r*CH_4_ value. The increase in the product yields was limited under the higher (>0.2) *r*CH_4_ condition. The yields of β-alanine (β-amino acid) and γ-aminobutyric acid (γ-amino acid) decreased at *r*CH_4_ > 0.3, which differed from the α-amino acid production scenario.

### 3.2. Formation of Carboxylic Acids

Figure 5 shows a total ion chromatogram of the products by spark discharge in a mixture of CO_2_, N_2_, and H_2_O. Here, the product was not hydrolyzed. A chromatogram of standard carboxylic acids is shown in Appendix A. Proton irradiation of mixtures of CH_4_, N_2_, and H_2_O, and H2O in various mixing ratios also produced similar chromatograms. The chromatogram shows the formation of various carboxylic acids via the proton irradiation of CO_2_ and H_2_O. When the products were alkaline-hydrolyzed, carboxylic acid yields did not increase, and sometimes decreased. This suggests that free carboxylic acids (not precursors) were mostly formed by proton irradiation. Figure 6 shows that the addition of CH_4_ increased the yield of carboxylic acid. Major products included formic acid (HCOOH; C_1_), acetic acid (CH_3_COOH; C_2_), and oxalic acid (HOOC-COOH; C_2_di); followed by monocarboxylic acids such as propionic acid (C_2_H_5_COOH; C_3_) and isolactic acid ((CH_3_)_2_CHCOOH; iC_4_); and dicarboxylic acids, such as malonic acid (HOOC-CH_2_-COOH; C_3_di) and succinic acid (HOOC-CH_2_-CH_2_-COOH; C_4_di).

The yields are also presented in Figure 6. We found lower yields of iC3, iC4, and C3di acids in the spark discharge experiments compared to the proton irradiation experiments (Figure 6), indicating different formation mechanisms between the two sets of experiments. However, the differences in yields are uncertain at this moment. The formation processes contributing to the differences in the molecular distributions will be discussed in detail in another paper.

## 4. Discussion

### 4.1. Formation of Amino Acids and Carboxylic Acids in Non-Reducing and Weakly Reducing Gas Mixtures

In the 1950s–70s, a wide variety of experiments on the abiotic synthesis of bioorganic compounds in strongly reducing gas CH_4_-NH_3_ mixtures by various energy sources detected amino acids [5,6,7,9] and nucleic acid bases [8,47]. They postulated that solar UV and lightning events were the major energy sources for prebiotic chemistry among various energy sources available on early Earth [16]. Recent studies suggest that the N_2_–CO_2_-rich atmosphere of the Hadean Earth contained a small amount of reducing carbon species (CH_4_ and/or CO) [15]. Thus, in the present experimental study, we used a mixture of N_2_, CO_2_, CH_4_, and H_2_O vapor.

A mixture of methane and nitrogen was used in several studies that characterized the impact of various energy sources. N_2_ is not dissociated by UV radiation with the wavelength > 110 nm [48], and, thus, solar UV could not provide efficient energy to synthesize amino acids in the early Earth’s atmosphere, even though its energy flux was much higher than energy fluxes from other sources. In contrast, extreme UV (EUV) emission is an efficient factor for nitrogen fixation, but it is absorbed in altitudes > 90 km above the ground, and thus can provide a significant amount of atomic nitrogen and other dissociated molecules required to produce HCN in the upper atmosphere [49]. However, efficient delivery of HCN to the lower atmospheric layers is problematic because of the slow vertical diffusion throughout the upper atmosphere.

Lightning flashes, which were estimated to have the second-largest energy flux [16], represent another energy source to synthesize amino acids in CH_4_–N_2_-type gas mixtures [17,50]. However, the rates of lightning flashes formed during thunderstorms were assumed to be comparable to the current rates observed over the land on Earth; the flash rates are lower by a factor of 30 over the ocean, which made up most of the area of the early Earth [51]. In addition, the flash rates could have been significantly lower given the lower atmospheric temperatures of the Hadean Earth under the faint, young Sun. This would have resulted in a drier atmosphere, with thus, reduced evaporation and associated thunderstorm formation, thunderstorm and associated lightning formation. The lightning flash rates may have been partially offset by the volcanic lightning resulting from the electrification of volcanic plumes [52]; however, its fraction could not have been high, as the continental fraction on the Hadean Earth was less than 1% [53].

The CH_4_–N_2_ gas mixture also yielded amino acids by proton irradiation, simulating the impact of GCRs [21]. These results suggested that spark discharges and proton irradiation could efficiently dissociate and ionize atmospheric N_2_ molecules to form N-containing organic compounds. CO_2_ and CO are not adequate starting materials for amino acid synthesis together with N_2_ by spark discharges, unless a large amount of H_2_ is added (*p*H_2_ > *p*CO or *p*CO_2_) to the gas mixtures [17].

The CO–N_2_ gas mixtures could yield amino acids by proton irradiation [21,22,24]. In the present study, we examined possible amino acid formation in the CO_2_-N_2_ gas mixtures with CH_4_ as an accessory component. As shown in Figure 3, amino acids were able to be formed by spark discharges when the CH_4_ ratio was 0.15 or over, but no detectable amino acids were obtained when the CH_4_ ratio was 0.1 or less. The presence of methane in the early Earth’s atmosphere was expected, but its mixing ratio was estimated to be much lower than 0.05 [54]. This is because there were no significant sources of methane on Earth other than volcanic activity, except for transient events associated with impacts, which could have supplied a significant fraction of the atmospheric methane for a few million years during the Hadean period [36]. Thus, the formation of amino acids by lightning flashes was not expected to be efficient on early Earth (ca. in the first half of a billion years). On the other hand, amino acid formation by proton irradiation via frequent SEPs was possible under the limited amount of CH_4_ that was present in the atmosphere of early Earth.

Carboxylic acids do not contain N atoms, which suggests that energy-expensive nitrogen fixation is not required for their formation, and, thus, it is easier to form them than amino acids. Garrison et al. reported that formic acid and formaldehyde were formed when an aqueous solution of CO_2_ and Fe^2+^ was irradiated with helium ions from an accelerator [55]. In the present study, various monocarboxylic acids (e.g., formic acid and acetic acid) and dicarboxylic acids (e.g., oxalic acid and malonic acid) were formed, both by spark discharge and by proton irradiation of the CO_2_–N_2_–H_2_O mixture. The yields of carboxylic acids generally increased when CH_4_ was added to the mixture, but did not change at CH_4_ mixing ratios from 0.05 to 0.25. Thus, the gas mixtures with low *r*CH_4_ (≤0.05) were still good potential starting materials to investigate the source of carboxylic acids.

### 4.2. Energy Yields of Amino Acids and Carboxylic Acids Formed by Spark Discharges and Proton Irradiation

Energy yields in radiation chemistry are usually expressed as the *G*-values, which is the number of molecules of product formed per 100 eV [56]. We calculated the *G*-values of glycine formed by spark discharges and by proton irradiation, which are shown in Figure 7. The *G*-values are presented in the logarithmic scale. Hereafter, *G*_Gly_ specifies the *G*-value of glycine.

The energy yield of glycine, *G*_Gly_, formed by proton irradiation was 0.047 at *r*CH_4_ = 0.5 (i.e., equimolar mixtures of CH_4_-N_2_ without CO_2_), which is about 2 orders higher than that obtained by spark discharges (0.00041). The difference between these two energy sources increases at low *r*CH_4_ values. For example, at *r*CH_4_ = 0.2, *G*_Gly_ by proton irradiation and spark discharges were 0.29 and 0.000059; the former is about 5000 times larger than the latter. At the lower *r*CH_4_ values, *G*_Gly_ by spark discharge was negligibly small.

Figure 8 shows the total estimated formation rate of glycine on Earth, assuming the energy flux of lightning flashes to be 1.0 × 10^24^ eV m^−2^ yr^−1^, as in Miller and Urey’s research [16]. As discussed above, the GCR particle flux around early Earth was 2 orders of magnitude lower than that currently observed, and is assumed to be 3 × 10^19^ eV m^−2^ yr^−1^ [22,48]. On the other hand, the annual SEP particle flux associated with superflares from the young Sun at 4–4.4 Ga was 5–7 orders of magnitude larger than the GCR flux [32]: the proton flux provided by SEP events from the young Sun during the early Hadean period is estimated to have been at least 3 × 10^24^ eV m^−2^ yr^−1^. This suggests the frequency of superflares with an energy of 10^34^ erg, as observed on an analog of the young Sun, EK Dra, to be about 100 events per year [28].

The formation rate was calculated by the following equation:Formation rate [kg yr^−1^] = *G*-value [molecules/100 eV] × Energy flux [eV m^−2^ yr^−1^]× Earth surface area [m^2^] × 0.0751 kg/mole × 100/(6.02 × 10^23^ molecules/mole) 


For CH_4_, as the major carbon species in the early Earth’s atmosphere, the estimated glycine production rate by lightning flashes was greater than that by GCRs and SEPs. For example, the largest moon of Saturn, Titan, has a CH_4_-N_2_ type atmosphere (CH_4_ >> CO_2_) [57]. Thus, in methane-dominated atmospheres, lightning flashes can be the major source of amino acid formation if the lightning frequency is high enough. On the other hand, at *r*CH_4_ < 0.3, the glycine production rate by lightning is estimated to be less than that by SEP events. For the weakly reduced volcanic atmosphere of early Earth, the methane molar ratio is expected to have been lower than 0.05 [47]. At *r*CH_4_ = 0.01, the production rates of glycine via GCRs and SEPs were ~10^3^ kg yr^−1^ and 10^8^ kg yr^−1^, respectively, while at *r*CH_4_ = 0.1, they increased to 10^6^ kg yr^−1^ and 10^11^ kg yr^−1^, respectively. The SEP-driven rates corresponded to 0.3 nmol cm^−2^ yr^−1^ and 300 nmol cm^−2^ yr^−1^, respectively, and were comparable to the lightning-driven production rates of amino acids derived by Stribling and Miller [19]. Their reported rate was 10 nmol cm^−2^ yr^−1^ for the early Earth’s hydrosphere with the reducing early Earth atmosphere (H_2_/C = 0.5–4 or *r*CH_4_ = 0.25–2). This suggests that under a weakly reducing (e.g., *r*CH_4_ = 0.01) early Earth atmosphere, SEPs could produce amino acid concentrations over one order of magnitude greater than those produced by lightning processes. Further molecular evolution toward the emergence of life would require high local concentration of the amino acids; thus, the pathway presented in this paper appears to be one of the key sources of the amino acid supply that was needed for the emergence of life.

These rates can also be compared to the amount of glycine delivered by extraterrestrial objects. Chyba and Sagan estimated that the organic carbon exogenous delivery rate 4 billion years ago was 10^6^ g-C yr^−1^ [58]. If we assume 2.0 % organic carbon and 0.0006% glycine, with about 0.06% of organic carbon contained in glycine (C_2_H_5_O_2_N), as measured from the Murchison meteorite as typical values for exogenous delivery [59], then the expected delivery rate of glycine to early Earth via carbonaceous chondrites is ~1 kg yr^−1^. Thus, the meteoritic amino acid delivery rates should have been greater than the endogenous production rates via “conventional” energy sources such as lightning and solar UV. However, if we include the contribution of SEP events to the production rates of the young Sun, then the endogenous amino acid production could have been much higher than exogenous delivery.

### 4.3. Where and How Life Originated?

Our results have broader implications as to not only when life originates, but also where and how. Our proton irradiation and spark discharge experiments, performed in weakly reducing gas mixtures resembling the early Earth’s atmosphere, demonstrated the production of amino acids and carboxylic acids, and thus emphasize the critical importance of atmospheric chemistry impacted by non-thermal energy sources as the major source of the precursors of life, as first suggested in [3,4,5,34]. Our study has important implications for the emergence of precursors of life in the early Hadean period of the Earth. This is consistent with recent studies suggesting that the basic conditions for the emergence of life were met as early as 4.4 billion years ago [60]. At this time, the Sun was a particularly magnetically active star, producing frequent and energetic superflares that could have been associated with fast CMEs and high-fluence SEP events.

Specifically, in the first 100 million years corresponding to the early Hadean period (4.4 Gyr ago), solar superflares (with an energy of >10^34^ erg) and associated SEPs occurred at a rate of one event per 3–10 days. The cumulative frequency of the occurrence of flares, *N* (*E* > *E*_0_), on the Sun and solar-like stars is universally scaled with flare energy, E, by a power law function, with an index of −1 [61]. By the end of the Hadean period (600 Myr old Sun), the observations of flares in solar-like stars of comparable age suggest that the frequency of occurrence decreased by a factor of 10. The rotation period of the Sun increased from about ~2–3 days for the 100-Myr-old Sun to 9–10 days for the 600-Myr-old Sun, which was caused by a loss of angular momentum in solar-like stars via magnetized stellar winds and coronal mass ejections [62]. Thus, as the frequency of the non-thermal energy input from the eruptive events into the planetary atmosphere decreased over the next few hundred million years, the production rate of complex molecules became comparable to or less than their destruction rate; thus, the role of SEP events became less important. This suggests that by understanding the time evolution of solar eruptive processes, we can refine the time window for the efficient formation of biologically important molecules, including amino acids, in the atmosphere of the early Earth by the young Sun. Specifically, it is likely that the window of SEP-efficient production of precursors of life ended in the late Hadean period (600 Mya).

Next, recent solar and stellar observations suggest that frequent, energetic superflares and associated coronal mass ejections from the young Sun produced high fluxes of SEPs, over 7 orders of magnitude higher than the GCR fluxes around the Earth in the early stage (the first 100 Myr) of the solar system [25,29,33,63]. Thus, GCRs appear to be less effective in the production of biomolecules than SEPs.

The energetic protons (with energy > 300 MeV) from such energetic SEP events would have reached the lower atmosphere of early Earth, forming air showers represented by extensive cascades of ionized particles and ionizing radiation that enhanced ionization via collisions with atmospheric species. These interactions formed a broad energy distribution of multiple cascades of electrons at energy > 35 eV, which subsequently thermalized to lower energies in the atmosphere via collisions. As soon as their energy reached 10 eV, they became efficient in breaking N_2_ bonds into odd nitrogen, along with the subsequent formation of NO_x_ and other molecules, including CO_2_, into CO and O [63]. For example, a single SEP proton, with energy of 500 MeV, can produce cascades of at least 10^7^ electrons. An energy level of 10 eV is required for nitrogen fixation, and this is the first step in achieving the reactive chemistry required to produce HCN, the major precursor molecule of prebiotic chemistry.

In contrast, the electric fields in spark discharges during lightning events produced ~100–200 keV electrons in the lower atmosphere of Earth, which would generate over 3 orders of magnitude fewer secondary cascades than a SEP proton [64]. This is consistent with our experimental findings, indicating that the production rate of amino acids produced by proton irradiation at 0.12 of the initial methane ratio, *r*CH_4_, was about 3 orders of magnitude greater than that produced by spark discharge. A typical single lightning bolt is only 2–3 cm wide and lasts for a fraction of a second, thus representing a local atmospheric event, while SEP precipitation occurs over the area of the planet’s polar cap, representing an open magnetospheric field controlled by the dynamic pressure of the young Sun’s wind and coronal mass ejections [34,65]. SEPs associated with superflares and coronal mass ejection events precipitated the atmosphere over an extended polar cap of the early Earth, thus representing a global contributor to atmospheric ionization that can last for hours (per SEP event) [34]. The formation of radicals (species with unpaired outer electrons) is mediated by 10 eV cascading electrons formed in the SEP-driven air showers. These involve reactions of nitric oxide with a highly reactive methylidyne radical (CH), which produced HCN as the organic feedstock of prebiotic chemistry in the early Earth’s atmosphere, as first suggested in [34].

Within our scenario, during the activity period of the young Sun, which lasted for about 600 million years (from 4.4 to 3.8 Gya), these precursors of life would have precipitated onto subaerial landscapes containing a variety of aqueous environments. These could have included freshwater subaerial hot springs [66]; hydrothermal lakes [67]; crater lakes [68,69]; Darwin’s “warm little ponds”, formed inside small volcanic islands similar to the Hawaiian island chain [70,71]; marine margins (beaches and lagoons) [72,73]; or arid intermountain valleys [74], as well as saline immersed settings such as submarine alkaline hydrothermal vents [75]. Which sites are the most favorable for the synthesis of amino acids and more complex molecules, including purines and pyrimidines?

First, the sizes and rates of dispersion and dilution of aqueous environments will affect the concentration of the delivered materials and their availability for further prebiotic reactions. This fact favors “little” ponds (per Darwin), and would, thus, rule out larger water bodies such as lakes or oceans. Thus, the atmospheric production of organic precursors is most beneficially delivered to these smaller, concentrated settings, but would not be able to participate in chemistry if deposited into larger lakes, rivers, or oceans, even if favorable energetic environments of alkaline hydrothermal vents were present.

Second, as discussed by Miyakawa et al. [76], HCN and formaldehyde cannot be efficiently polymerized in warm waters (>30 °C). Thus, hot springs and hydrothermal lakes would not be the most favorable environments for the accumulation of highly volatile HCN molecules. The produced molecules precipitated down into these low-temperature environments, where they could be efficiently accumulated and produce simple prebiotic compounds including amino acids, nucleobases, and other biologically relevant molecules [70,71]. These prebiotic molecules were subject to wet–dry cycles as plausible drivers of prebiotically important reactions, such as polymerization, in a regular periodic timeframe of minutes to hours, or, at most, days, rather than seasonally [67,71,77,78,79]. The synthesized amino acids would likely be polymerized via condensation reactions forming peptides [80]. It is worth mentioning that if carboxylic acids are deposited on mineral surfaces, they can form dried reactive films which then assemble into membranous vesicles [81].

Third, submarine alkaline hydrothermal vents could also represent promising environments for prebiotic chemistry because of the reduced condition, continuous supply of thermal energy, and the presence of a high concentration of transition-metal ions as catalysts [82]. Kurihara et al. [83] have experimentally demonstrated the formation of organic aggregates, possible precursors for primitive cellular structures, from an aqueous solution of abiotically formed complex organic compounds produced via the irradiation of high-energy protons (simulating SEP-driven energy input into the N_2_–CO_2_ rich atmosphere) in a hot flow reactor simulating the flow of hydrothermal vent environments. Thus, early Earth had a diversity of environments that was favorable for the precipitation of atmospherically produced precursor molecules and subsequent prebiotic synthesis, increasing the chances that the chain of processes producing the first cell-like structures would be completed.

A continuous reaction network driven by persistent non-thermal energy sources from SEPs in Earth’s first 600 million years could have contributed to the development of the chemical complexity that would have subsequently produced RNA precursors, and, ultimately, molecules with the properties of information storage and replication following natural selection, or a primordial RNA world [79,84].

## 5. Conclusions

We have, for the first time, experimentally shown that the production rates of amino acids and carboxylic acids in non-reducing gas mixtures (i.e., CO_2_–N_2_ mixture without any reducing carbon sources) due to proton irradiation can significantly exceed the production rates of these molecules via GCRs and spark discharges. This provides experimental evidence supporting the importance of SEP events in the young Sun as energy sources which were required for the synthesis of the biologically important molecules deposited and accumulated in diverse aquatic geological settings of the early Earth, as suggested in [34]. A continuous reaction network driven by persistent non-thermal energy sources from SEPs in Earth’s first 600 million years could have contributed to the development of the chemical complexity that would have subsequently produced RNA precursors, and, ultimately, the molecules with the properties of information storage and replication following natural selection, or a primordial RNA world [79,84].

Our experimental results also suggest that endogenous production of amino acids on Earth via SEPs could have surpassed that of extraterrestrial delivery via impacts from comets and carbonaceous chondrites [58,59].

The present study reveals an exciting possibility that the SEP-driven mechanism of nitrogen fixation could have been efficient in the production of amino acids, carboxylic acids, and other biomolecules. This is not only the case on early Earth, but possibly on Mars, as implied by the recent discovery of abundant (up to 1100 ppmv) nitrates on its surface by the Curiosity Rover, suggesting efficient production of odd nitrogen in the Martian atmosphere [85].

## Figures and Tables

**Figure 1 life-13-01103-f001:**
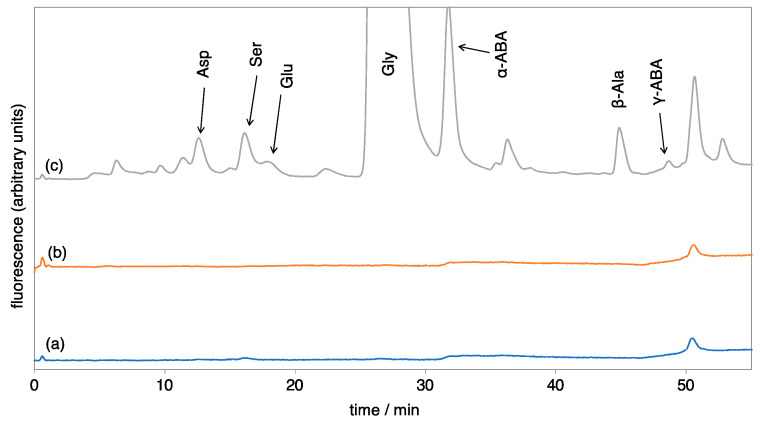
HPLC chromatograms of amino acids formed by spark discharges in gas mixtures of CO_2_, CH_4_, N_2_, and H_2_O in various mixing ratios. (a) *r*CH_4_ = 0, (b) *r*CH_4_ = 0.05, (c) *r*CH_4_ = 0.5. All the samples were analyzed after acid hydrolysis. Compositions are shown in Table 1. ABA means aminobutyric acid.

**Figure 2 life-13-01103-f002:**
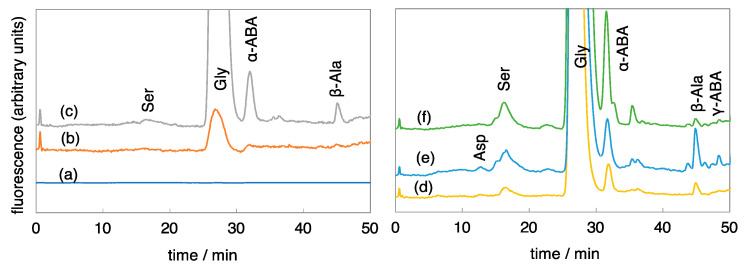
HPLC chromatograms of amino acids formed by proton irradiation of gas mixtures of CO_2_, CH_4_, N_2_, and H_2_O in various mixing ratios. (a) *r*CH_4_ = 0, (b) *r*CH_4_ = 0.01, (c) *r*CH_4_ = 0.02, (d) *r*CH_4_ = 0.05, (e) *r*CH_4_ = 0.2, (f) *r*CH_4_ = 0.5. All the samples were analyzed after acid hydrolysis. Compositions are shown in Table 1. ABA means aminobutyric acid.

**Figure 3 life-13-01103-f003:**
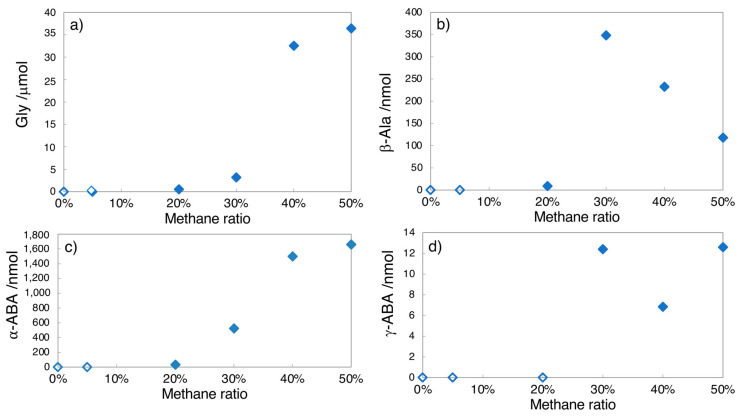
Dependence of amino acid yields by spark discharges on methane ratio (*r*CH_4_). (**a**) Glycine, (**b**) β-alanine, (**c**) α-amino butyric acid, (**d**) γ-aminobutyric acid. All the samples were analyzed after acid hydrolysis. Open symbols show “not detected”.

**Figure 4 life-13-01103-f004:**
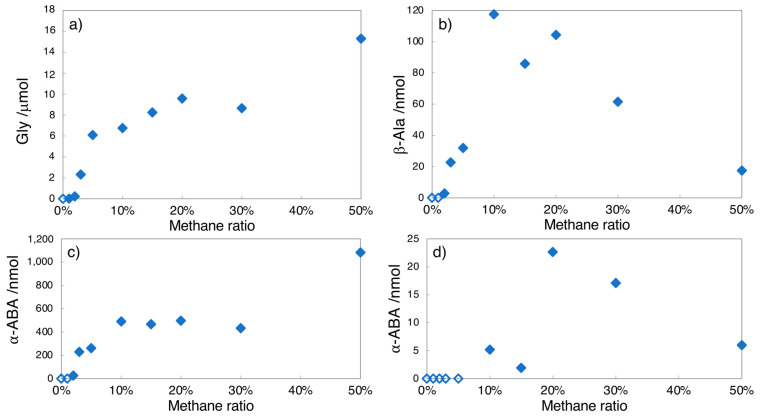
Dependence of amino acid yields by proton irradiation on methane ratio (*r*CH_4_). (**a**) Glycine, (**b**) β-alanine, (**c**) α-amino butyric acid, (**d**) γ-aminobutyric acid. All the samples were analyzed after acid hydrolysis. Open symbols show “not detected”.

**Figure 5 life-13-01103-f005:**
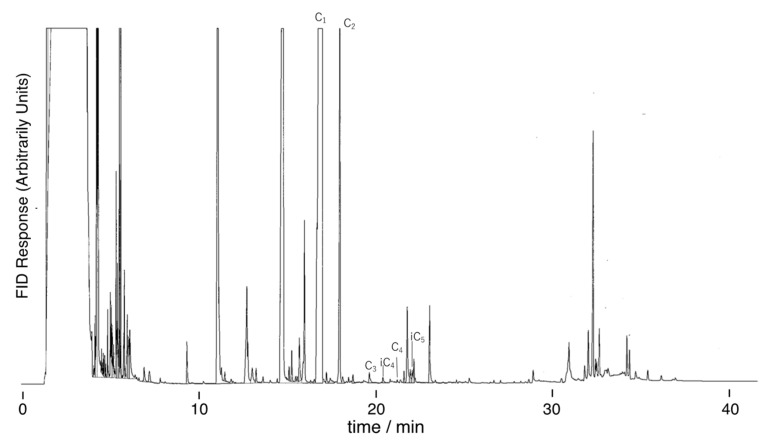
GC chromatogram of carboxylic acids formed by spark discharge in the gas mixture of CO_2_, CH_4_, N_2_, and H_2_O (*r*CH_4_ = 0.25). The sample was analyzed without hydrolysis. C_1_: formic acid, C_2_: acetic acid, C_3_: propanoic acid, iC_4_: isobutyric acid, C_4_: butyric acid, iC_5_: isovaleric acid.

**Figure 6 life-13-01103-f006:**
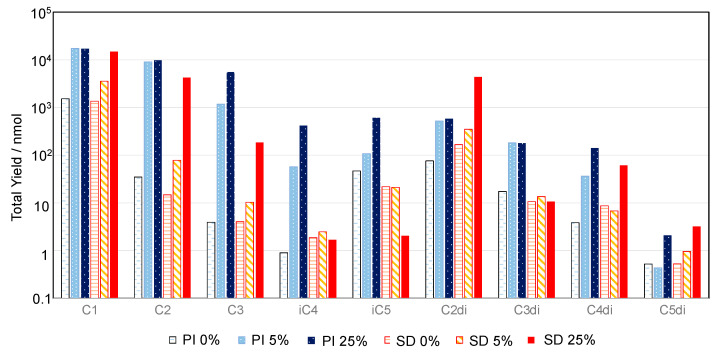
The yields of carboxylic acids formed from the gas mixtures of CO_2_, CH_4_, N_2_, and H_2_O by spark discharges (SD) or proton irradiation (PI). The samples were analyzed without hydrolysis. The methane ratio (*r*CH_4_) was 0%, 5%, or 25%, as indicated in the figure. C_1_: formic acid, C_2_: acetic acid, C_3_: propanoic acid, iC_4_: isobutyric acid, iC_5_: isovaleric acid, C_2_di: oxalic acid, C_3_di: malonic acid, C_4_di: succinic acid, C_5_di: glutaric acid.

**Figure 7 life-13-01103-f007:**
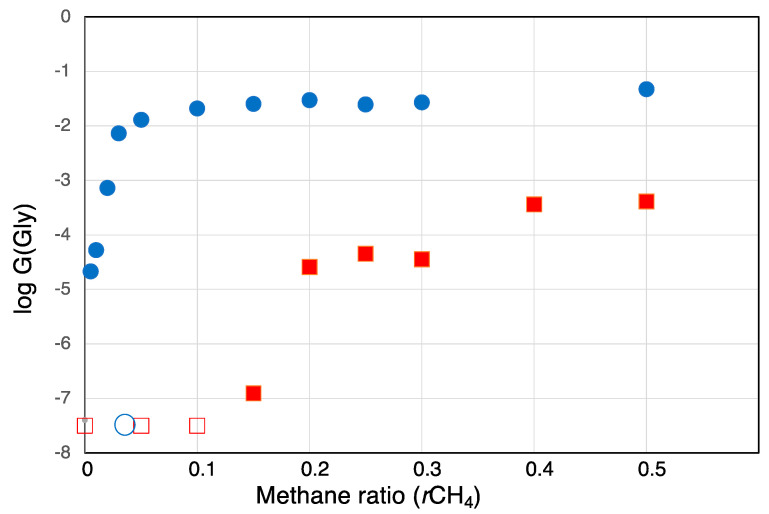
The energy yields (*G*-values) of glycine formed from the gas mixtures of CO_2_, CH_4_, N_2_, and H_2_O by spark discharges and proton irradiation as a function of *r*CH_4_. Circle: proton irradiation, square: spark discharge, open symbol: not calculated (*G* was less than 10^−7^).

**Figure 8 life-13-01103-f008:**
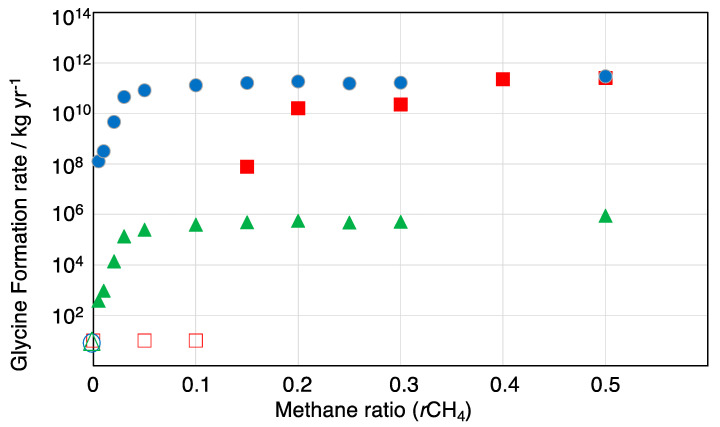
The estimated production rate of glycine by spark discharge (lightning), GCRs, and SEP events in the CO_2_-, CH_4_-, N_2_-, and H_2_O-dominated atmosphere of early Earth. Circle: SEPs, triangle: GCRs, square: gas discharge, open symbol: not calculated (less than 10^2^ kg yr^−1^).

**Table 1 life-13-01103-t001:** Composition of starting materials.

CH_4_ Ratio (*r*CH_4_)	Pressure (*p*) *			Volume
	CH_4_	CO_2_	N_2_	H_2_O
	(Torr)	(Torr)	(Torr)	(mL)
0%	0	350	350	5
0.5% **	3.5	346.5	350	5
1% **	7	343	350	5
2% **	14	336	350	5
3% **	21	329	350	5
5%	35	315	350	5
10%	70	280	350	5
15%	105	245	350	5
20%	140	210	350	5
25%	175	175	350	5
30%	210	140	350	5
40% ***	280	70	350	5
50%	350	0	350	5

* CH_4_ ratio (*r*CH_4_) = *p*CH_4_/(*p*CH_4_ + *p*CO_2_ + *p*N_2_). ** Proton irradiation only; *** spark discharge only.

## Data Availability

Not applicable.

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
