# Peer review of "Formation of Amino Acids and Carboxylic Acids in Weakly Reducing Planetary Atmospheres by Solar Energetic Particles from the Young Sun"

_life, 2023, doi:10.3390/life13051103_

Round 1
Reviewer 1 Report
See report attached.

Author Response
Dear Reviewer 1
Please see the attached file.

Reviewer 2 Report
I have no expertise whatsoever on experimental high energy chemistry, but I trust the Editor has also sought the opinion of other referees who do.
I do have reasonable expertise in solar/stellar activity, flares, and reconstruction of activity in the "young sun", which is the primary motivation for the experiments reported upon in this paper. My report is thus restricted to these aspects. In fact the case for "superflares" from the young sun may be even more robust than suggested by the literature cited in the introductory section.
A first important point is that there is likely nothing fundamentally special
about "superflares"; they likely represent the high end of a scale invariant universal distribution. In the case of the sun, it is long known (e.g., Dennis 1985, Solar Physics, vol. 100, 465) that the frequency distribution of peak X-Ray flux is described by the same power-law at all phases of the solar activity cycle, only the overall frequency waxes and wanes in the course of the cycle. Moreover, similar scale-free frequency distributions seems to characterize both the sun and far more active stars (e.g. Audard et al. 2000, The Astrophysical Journal, vol. 541(1), 396; although this notion has not gone unchallenged, see, e.g., Sakurai 2022, Physics, vol. 5(1), 11-23)
Another important point is that the exact form of the relationship between X-Ray emission and SEP flux in solar flares, and extension of such relationships to stellar flares, remains highly uncertain. For example, the recent paper by Herbst et al (2019, Astronomy & Astrophysics, vol. 621, A67) suggests that current extrapolations of SEP fluxes to very activity regimes such as observed in young dwarf stars may be off by up to five orders of magnitude, no less! The proton intensities used in the experiments reported upon in this paper (section 2-3) may not be so unrealistically high after all.
Finally, inferring flaring activity of the young sun from observation of cooler, more active dwarfs may seem to be a stretch, but observations do suggest that independently of spectral type, late-type stars behave similarly in terms of the observed relationship between surface magnetism and X-ray emission, versus rotation rate (as quantified by the convective Rossby number; see Wright et al 2018, Monthly Notices of the Royal Astronomical Society, vol. 479, 2351; Reiners et al. 2022, Astronomy & Astrophysics, vol. 662, A41).
Getting into these in all detail in the introduction would clearly be overkill and too much of a distraction, but adding a few sentences and references to better buttress the (indirect) observational support for enhanced SEP fluxes from the young Sun might further strengthen the motivation for the paper. I leave it to the authors to decide if and at what level they wish to do so.
Being unfamiliar with these types of chemical analyses, I was wondering if measurement uncertainties can be indicated as error bars on Figs 3, 4 7 and 8; or are the laboratory measurements so precise that error bars are smaller than the plotted symbols ?
Regarding the discussion (section 4.1), I was wondering if lightning associated with enhanced volcanic activity in the Haedan period could raise lightning rate sufficient to have a significant effect on amino acid generation?
On p 10, line 4, the age of "early Earth" is gven as "ca. 4Ga". Presuming what is meant here is 4 Gigayears, this would place this early Earth already quite late in the spin-down of the Sun, with overall magnetic/flaring activity not so much more intense than the present-day Sun; or is this supposed to mean 4Gy into the past, so a truly early Earth aged ~0.5 Gyr ?
Typo/grammar:
p2. 3rd paragraph, 3 lines from bottom: "...magnitude greater..."
p9, section 4.1, second par.: "...is no dissociated by..."
p9, section 4.1, second par.: "Thundering was estimated..."
p11 4 lines from bottom: "...molar ratio..." (I think)
Author Response
Dear Reviewer 2
Please see the attached file.
